# Solvent tuning of photochemistry upon excited-state symmetry breaking

Bogdan Dereka[1,2], Denis Svechkarev[3], Arnulf Rosspeintner[1], Alexander Aster[1], Markus Lunzer [4], Robert Liska[4], Aaron M. Mohs[3,5] & Eric Vauthey [1✉]

The nature of the electronic excited state of many symmetric multibranched donor–acceptor molecules varies from delocalized/multipolar to localized/dipolar depending on the environment. Solvent-driven localization breaks the symmetry and traps the exciton in one branch. Using a combination of ultrafast spectroscopies, we investigate how such excited-state symmetry breaking affects the photochemical reactivity of quadrupolar and octupolar A–($\pi$-D)$_{2,3}$ molecules with photoisomerizable A–$\pi$–D branches. Excited-state symmetry breaking is identified by monitoring several spectroscopic signatures of the multipolar delocalized exciton, including the S$_2 \leftarrow$ S$_1$ electronic transition, whose energy reflects interbranch coupling. It occurs in all but nonpolar solvents. In polar media, it is rapidly followed by an alkyne–allene isomerization of the excited branch. In nonpolar solvents, slow and reversible isomerization corresponding to chemically-driven symmetry breaking, is observed. These findings reveal that the photoreactivity of large conjugated molecules can be tuned by controlling the localization of the excitation.

[1] Department of Physical Chemistry, University of Geneva, 30 Quai Ernest-Ansermet, 1211 Geneva, Switzerland. [2] Department of Chemistry and Institute for Biophysical Dynamics, James Franck Institute, The University of Chicago, 929 E. 57th St., Chicago, IL 60637, USA. [3] Department of Pharmaceutical Sciences, University of Nebraska Medical Center, Omaha, NE 68198-6858, USA. [4] Institute of Applied Synthetic Chemistry, TU Wien, Getreidemarkt 9/163/MC, 1060 Vienna, Austria. [5] Department of Biochemistry and Molecular Biology, Fred and Pamela Buffett Cancer Center, University of Nebraska Medical Center, Omaha, NE 68198-6858, USA. ✉email: eric.vauthey@unige.ch

The current demand for highly efficient light-harvesting and photoresponsive materials is stimulating intense efforts in the design and synthesis of photoactive molecular architectures. Engineered multichromophoric arrays are emerging as promising photonic, photovoltaic and photocatalytic materials[1–3]. Significant work is being invested to understand, influence, and ultimately control exciton (de)localization in synthetic materials[3–6], conjugated polymers[7–10] and natural light-harvesting complexes[11–13]. Multipolar charge-transfer scaffolds represent one type of materials featuring strong interaction between D–$\pi$–A branches, D and A being electron donor and acceptor subunits, arranged in well-defined architectures. Exhibiting advantageous two-photon absorption (TPA) properties[14,15], they attract attention in various fields, including 3D nanofabrication[16–18], in vivo fluorescence microscopy[19–22], power limiting[23,24] and optical uncaging[25] among others[26–29].

Photoexcitation of these molecules leads to a delocalized and symmetric multipolar exciton that can undergo symmetry breaking due to surrounding solvent fluctuations[30–32]. Excited-state symmetry breaking (ES-SB) can be viewed as the decoherence of a multipolar exciton evenly delocalized over all branches of the molecule that eventually results in its confinement on a single branch. This phenomenon has seen a growing interest over the past few years, and was extensively investigated in linear quadrupolar rod-like molecules[30–44]. Attempts to identify a spectroscopic signature of ES-SB using electronic transient absorption in the UV–Vis region failed, because localized and delocalized states exhibit similar spectra[39,45–48]. However, broadband time-resolved fluorescence can follow the decrease of the emission dipole moment that occurs upon ES-SB[36,49]. The most direct real-time visualization of ES-SB can be achieved by monitoring vibrational modes localized at the ends or within the D–$\pi$–A branches of these systems, using ultrafast time-resolved infrared spectroscopy (TRIR). These studies revealed how the occurrence and extent of ES-SB depend on properties of the dye itself[30,32,34,35] and of its environment[30–33]. Until now, such TRIR investigations were limited to quadrupolar molecules.

We exploit this powerful spectroscopic toolbox to investigate how the (de)localization, (a)symmetry and (de)coherence of excitons affect the photochemical reactivity and functional symmetry[50]. We report on our investigation of the excited-state dynamics and photochemistry of octupolar (**O**) and quadrupolar (**Q**) dyes consisting of a triazine core decorated with dialkylanilines connected through alkyne $\pi$-bridges (Fig. 1a). The single-branch dipolar analogue (**D**) was recently shown to undergo an ultrafast alkyne–allene photoisomerization[51]. The resulting allene state is characterized by a strong charge-transfer character and orthogonal orientation of the triazine and aniline moieties that can be described as a twisted and rehybridized intramolecular charge-transfer (TRICT) state. Here, we investigate how the presence or absence of ES-SB affects this phototransformation in the multibranched **O** and **Q**, and its implications on the accessible photochemical pathways. By applying a large variety of ultrafast spectroscopic techniques, we show that, in all polar solvents, ES-SB leads to a fast concentration of the excitation on a single arm, enabling efficient and irreversible alkyne–allene photoisomerization in both **O** and **Q**. ES-SB is thus a key process that controls the photochemical reactivity of multipolar systems. In non-polar media, solvent-driven symmetry breaking is not operative, and the excitation density on each branch is too weak to make the alkyne–allene transformation efficient. This reaction occurs slowly and reversibly in one of the branches that is randomly selected by instantaneous solvent forces and structural fluctuations. This results in a transient symmetry-broken state that is not stable and quickly converts back to the initial state. To the best of our knowledge, this finding can be viewed as the first observation of a photochemically-driven symmetry breaking process. The results also demonstrate that the delocalization of the excitation leads to local changes of electronic distribution that are too weak to favor efficient photochemistry, and thus does not result in qualitatively different photochemical pathways that would involve its inherent coherent nature and lead to functional symmetry[50].

## Results

**Basic photophysical properties**. The relative intensities of the $S_1 \leftarrow S_0$ and $S_2 \leftarrow S_0$ bands in the one-photon (OPA) and two-photon (TPA) absorption spectra of **O** and **Q** point to mutually exclusive selection rules for OPA and TPA (Fig. 1b): the TPA spectra peak at higher frequencies featuring only weak shoulders at their corresponding one-photon maxima. These spectra can be explained within the simple Kasha excitonic model that considers the dipolar interactions between the three (two) D–$\pi$–A branches in **O** (**Q**) and leads to a specific splitting pattern of the excited states[52]. As depicted in Fig. 1c, the $S_2 \leftarrow S_1$ energy gap corresponds to the Davydov splitting and is directly related to the electronic coupling between the branches, V. It amounts to 2630 and 1700 cm$^{-1}$ for **O** and **Q** and gives very similar V values (880 and 850 cm$^{-1}$, respectively). The mutually exclusive selection rules for OPA and TPA observed with **O** and **Q** (Fig. 1c) are well reproduced by the Kasha model, indicating that both molecules have symmetric and multipolar ground state and Franck–Condon excited states. Despite this symmetry, the absorption spectra of **O** and **Q** exhibit a significant solvatochromism. For example, the $S_1 \leftarrow S_0$ band of **O** undergoes a ~2040 cm$^{-1}$ redshift by going from the non-polar cyclohexane to the highly polar DMSO. This solvatochromism can be explained by considering that **O** and **Q** cannot be described as point octupole or point quadrupole, respectively (Supplementary Figs. 1–3, Supplementary Tables 1–5).

A much stronger solvent dependence is observed with fluorescence, which shifts over the whole visible region, i.e. by about 9000 cm$^{-1}$, upon increasing solvent polarity (Supplementary Fig. 4). Along with this downshift, the fluorescence quantum yield decreases from ~0.4 to <10$^{-3}$ (Supplementary Table 6). The latter effect is due to both an increase of the non-radiative rate constant and a decrease of the transition dipole for emission. These observations are indicative of ES-SB in polar solvents. The strong solvatochromism points to a dipolar excited state, whereas the decrease of transition dipole moment is consistent with a localization of the excitation, i.e. with the decoherence of the exciton[36]. The Ivanov symmetry-breaking model[53–55] can be applied to evaluate the dipolar character, D, of the relaxed $S_1$ state. It is based on two experimentally accessible parameters: the electronic interbranch coupling, V, and the fluorosolvatochromic shift in a given medium with respect to a non-polar solvent. This model predicts complete exciton localization even in weakly polar media (D = 0.985 in benzene and D = 0.993 in chloroform, with D = 1 being a full localization on a single branch). This result differs from previous investigations, where complete localization was only achieved in the most polar solvents[30,34], or could not be realized at all[32,33]. It is possible here because of the weak electronic coupling between the branches (~850–900 cm$^{-1}$) and the large charge transfer in each arm upon photoexcitation.

**Excited-state symmetry breaking in the multibranched dyes**. To track the ES-SB dynamics in real time, ultrafast time-resolved infrared spectroscopy was carried out on C≡C local vibrational tags[30,34,35]. Figure 2a–c reveals that the spectra measured with **O** and **Q** are dominated by an excited-state absorption (ESA) feature that is much broader than the 1200 cm$^{-1}$ spectral window. The negative band around 2190 cm$^{-1}$ is a ground-state bleach

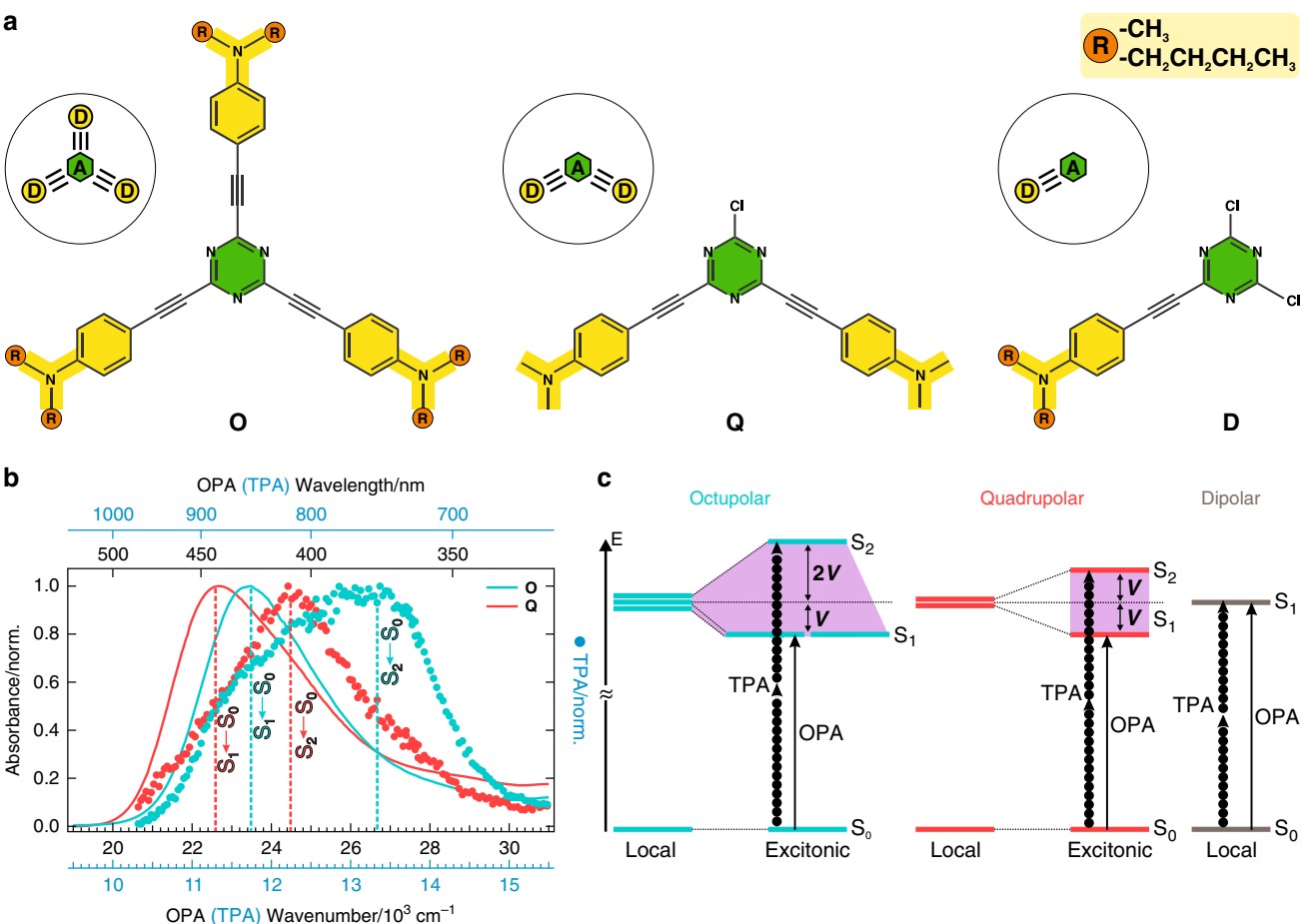

**Fig. 1 Multipolar compounds and their absorption spectra interpreted with the Kasha excitonic model. a** Structures of the triazine-based multipolar dyes **O** and **Q** along with the model dipolar analogue **D**. The electron-donating moieties (**D**) are in yellow and the acceptor (**A**) is in green, whereas R (orange) represents alkyl side chains. The butyl R groups make **O** and **D** soluble in alkanes. **b** One-photon (OPA, solid line) and two-photon (TPA, circles) absorption spectra of **O** (blue) and **Q** (red) in chloroform. TPA spectra correspond to blue axes, whereas OPA spectra are plotted on black axes, which are twice higher in energy. Dashed lines pinpoint the maxima of the dominant transition. **c** Illustration of the excitonic splitting for octupolar and quadrupolar systems within the Kasha exciton model, with the relation between the Davydov splitting (purple shading) and the electronic interbranch coupling, $V$, and the one-(solid arrows) and two-photon-allowed (arrows made of circles) transitions. The energy-level scheme of **D** is shown for comparison.

(GSB) feature and is also present in the stationary IR absorption spectrum (gray-shaded) (Supplementary Figs. 5, 6). Although the mid-IR spectral range usually reports on molecular vibrational transitions, the broad ESA band is assigned to an electronic transition of **O** and **Q**, namely the excitonic $S_2 \leftarrow S_1$ transition. In the Kasha model, the energy of this transition corresponds to the Davydov splitting, which is itself a direct consequence of the exciton delocalization. Therefore, the presence of this band at the earliest time delay (~200 fs) is an unambiguous signature of the delocalized and symmetric nature of the initial exciton that is spread over the entire molecule, whether octupolar or quadrupolar. This is confirmed by the absence of this band in the TRIR spectra measured with **D**, where excitation is confined on a single branch (Fig. 2d).

The dynamics of this excitonic band in non-polar and polar media are completely different. In the non-polar cyclohexane, this band is present over the entire 2 ns time window of the experiment and exhibits a bimodal decay on the 50 ps and ~1.5 ns timescales (Fig. 2a). Such slow dynamics evidence the preservation of the delocalized and symmetric nature of the exciton. In contrast, the decay of this band is ultrafast in all polar solvents, including weakly polar ones, and occurs on the hundreds of fs to few ps timescale. Afterwards, only weaker

and narrower vibrational bands are left in the 1900–2200 cm$^{-1}$ region (Fig. 2b, c, Supplementary Fig. 7).

The disappearance of this $S_2 \leftarrow S_1$ band reflects the loss of interbranch coupling as a consequence of the localization of the excitation on a single branch. This is a direct spectroscopic manifestation of the decoherence of the initially prepared exciton. After localization, the $S_1$ and $S_2$ states are of a completely different nature and should be similar to those of **D**. Therefore, this mid-IR excitonic band is a direct electronic signature of the delocalized exciton and its disappearance is a manifestation of ES-SB.

This assignment is supported by broadband fluorescence up-conversion spectroscopy (FLUPS) measurements of the multipolar dyes in non-polar and polar solvents. The fluorescence of **O** in non-polar solvents does not exhibit any significant spectral dynamics (Supplementary Fig. 27) and its intensity follows the same decay as the excitonic TRIR band (Fig. 3a). By contrast, in polar solvents, the fluorescence spectrum shifts to longer wavelength, and its intensity decreases to 1/3 (**O**) and 1/2 (**Q**) of its initial value on the same timescale as the decay of the excitonic TRIR band (Fig. 3b, c, Supplementary Figs. 25, 26, Supplementary Table 11). This result is fully consistent with a decrease of the transition dipole moment for emission of **O** and **Q** upon collapse of the exciton onto a single branch. The 2/3 and 1/2

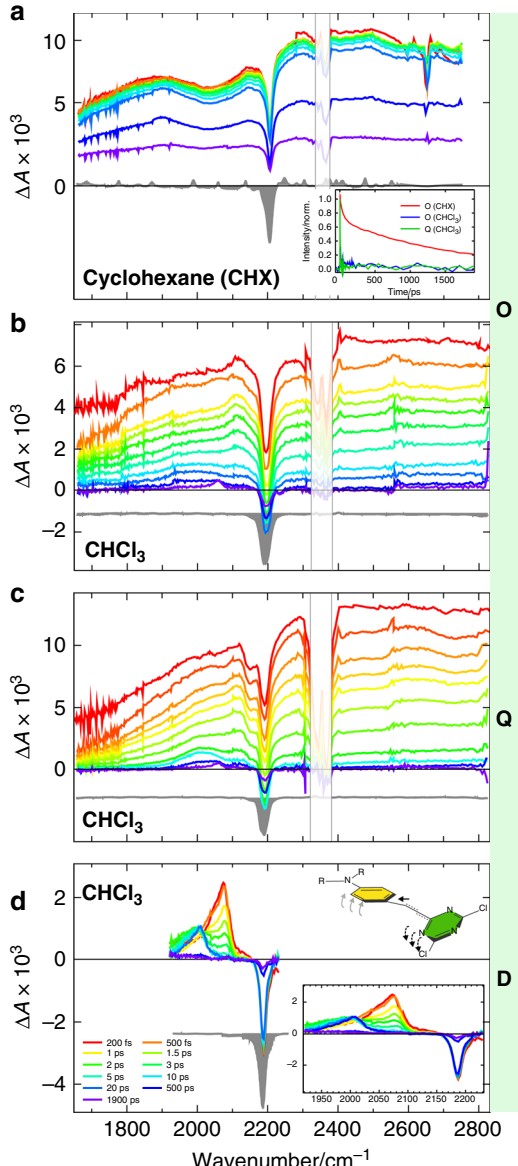

**Fig. 2 TRIR spectral evolution of the triazine-based systems upon $S_1 \leftarrow S_0$ excitation. a O** in the non-polar cyclohexane. **b O** in medium-polar chloroform and **c Q** in medium-polar chloroform. The inset in **a** shows the time dependence of the integrated 2400–2800 cm$^{-1}$ intensity for **a–c**. **d D** in the medium-polar chloroform showcasing the alkyne–allene transformation. The cartoon depicts the molecular motion associated with this photoisomerization (electron-accepting moiety is in green, electron donor is in yellow). The inset in **d** zooms into the spectral region with non-zero signal for **D**. The gray-shaded spectra are the negative stationary IR spectra (offset for clarity). The region near 2350 cm$^{-1}$ is shaded to cover interferences caused by atmospheric $CO_2$.

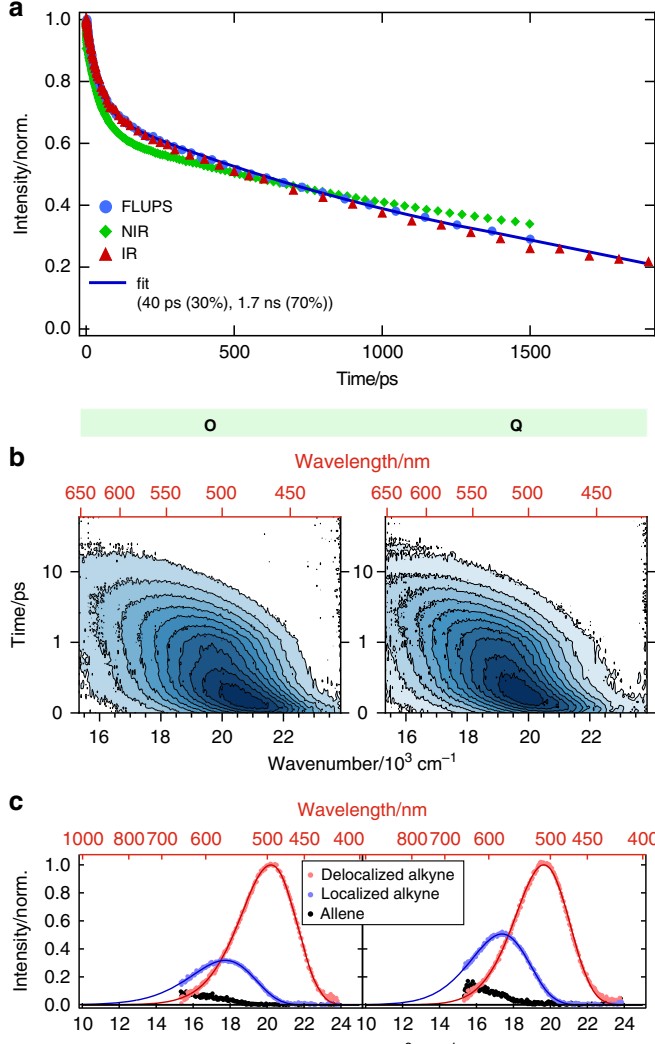

**Fig. 3 Delocalized exciton evolution and time-resolved fluorescence spectroscopy. a** Time evolution of the delocalized exciton **O** in the non-polar cyclohexane as extracted from three different methods: Fluorescence up-conversion spectroscopy (FLUPS) emission (blue circles), IR excitonic $S_2 \leftarrow S_1$ transition (red triangles) and higher-energy NIR transition (peaking at 863 nm, green diamonds). The solid line is a biexponential fit to the FLUPS data shown with the time constants and their respective amplitudes. **b** FLUPS contour plots obtained for **O** and **Q** in the highly polar benzonitrile. The equidistant blue-shaded contours are drawn every 10% from zero (white) to the maximum intensity (dark blue). The time axis is linear between 0 and 1 ps and logarithmic afterwards. **c** Results of a global analysis of the FLUPS data according to a consecutive A → B → C → scheme representing the emission of three distinct species/states: delocalized unrelaxed exciton (red), localized exciton (blue) and allene photoproduct (black). *Circles* represent experimental data points, solid lines are lognormal fits. The fluorescence intensity of the localized exciton is 1/3 and 1/2 of the delocalized one for **O** and **Q** correspondingly.

decreases of the fluorescence intensity are those expected for a transition from a state with the excitation delocalized over 3 or 2 branches to a state with the excitation confined on a single branch[36].

ES-SB occurs with **O** and **Q** in all polar solvents (Supplementary Discussion), including the weakly polar ethers or the quadrupolar benzene, as predicted by the Ivanov model[53–55]. Its dynamics are the same for both **O** and **Q** and depend only on the solvent. This confirms the results of previous investigations that solvent motion dictates the dynamics of ES-SB[30,32,33,35]. Global analysis of the TRIR data indicates that this process occurs

on the same timescale as solvation and can in general be reproduced with the sum of two exponential functions approximating the bimodal nature of solvation dynamics (Supplementary Table 7). The faster component is characteristic of the inertial solvent motion, whereas the slower one reports on diffusive solvation dynamics[56]. The contribution of inertial motion to ES-SB increases with solvent polarity. In the most polar media, e.g. DMSO, inertial solvation is essentially

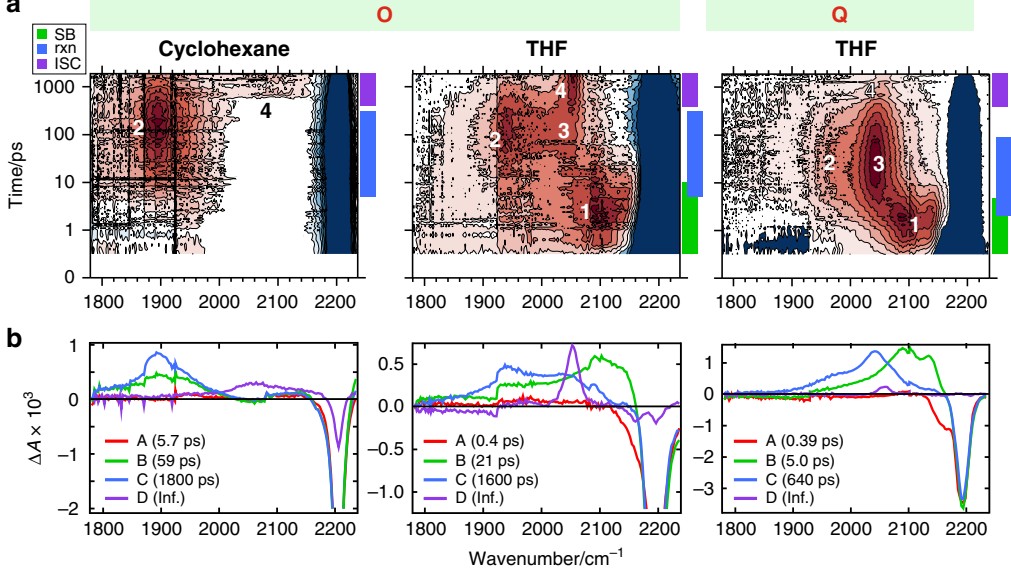

**Fig. 4 Double-difference TRIR spectroscopy of the multipolar compounds. a** Vibrational double-difference TRIR contour maps of **O** in the non-polar cyclohexane and of **O** and **Q** in the polar tetrahydrofuran (THF). Excited-state absorption bands are in red and marked with numbers (see text), ground-state bleach is in blue. The time axis is linear between 0 and 1 ps and logarithmic afterwards. Color bars indicate the dominant processes taking place over the specified timespan: symmetry breaking (SB, green), reaction (rxn, blue) or intersystem crossing (ISC, purple). **b** Evolution-associated double-difference spectra obtained from global analysis of the corresponding data shown in **a** assuming a consecutive A → B → C → D → reaction scheme.

responsible for the complete disappearance of the excitonic band. Even partial relaxation of these highly polar solvents is sufficient to trap the excitation on a single branch. In less polar solvents, complete diffusive reorientation is needed to achieve enough asymmetry in the solvent field to localize the exciton.

**Photoreactivity of the D–π–A branches.** Knowing that ES–SB occurs in a few ps with both molecules in all polar solvents, we now consider how this process affects the photoreactivity of the D–π–A branches. As shown previously[51], photoexcitation of the single-branch **D** results in the isomerization of the alkyne π-bridge absorbing at 2071 $cm^{-1}$ to an orthogonally twisted allene fragment with a characteristic absorption band at 1925 $cm^{-1}$ (Fig. 2d, Supplementary Figs. 8–13, Supplementary Table 8)[51]. This reaction, observed in polar as well as non-polar solvents, changes both the electronic properties of the bridge (alkyne → allene) and its geometry (planar → perpendicular), and leads to a local rehybridization ($sp$-linear → $sp^2$-bent) (Fig. 2d). This alkyne–allene isomerization is triggered by the large charge-transfer character of the alkyne excited state, and is accompanied by close to full intramolecular charge separation towards a TRICT state.

To access the structural dynamics of the π-bridges in **O** and **Q** directly after photoexcitation, the contribution of the broad excitonic ESA band was removed from the TRIR data. As detailed in Supplementary Discussion, this was done by subtracting all ESA bands (electronic and vibrational) present at the earliest time delay from the TRIR data after intensity scaling. For this reason, both the excitonic ESA band and vibrational ESA bands representing the initially populated excited state are not visible in the resulting double-difference spectra (Fig. 4 (top)). These spectra show changes relative to the earliest spectrum corresponding to the symmetric multipolar exciton.

For **O** in non-polar solvents, where symmetry is preserved during the whole excited-state lifetime, a single new ESA band (band 2) emerges at 1900 $cm^{-1}$ on 5.7 and 59 ps timescales (Fig. 4 (bottom), Supplementary Fig. 14). This frequency is characteristic of the allene stretching vibration. Comparison with the FLUPS

data reveals that the rise of band 2 occurs concurrently with a partial decay (~40%) of the symmetric excited-state population (Fig. 3a). Afterward, band 2 decays on the same 1.5–2 ns timescale as the residual fluorescence. The coexistence of the fluorescence and allene band 2 and their parallel decay suggest an equilibrium between the symmetric excited state and the allene state. This decay leads to the appearance of a new ESA band at ~2064 $cm^{-1}$ (band 4) also observed with **D** and assigned to the triplet excited state $T_1$. The associated GSB corresponds to ~20% of its initial value, close to that of 22% measured with **D** (Supplementary Table 9). The identical bleach amplitudes suggest a delocalized $T_1$ state in **O**. If $T_1$ were localized on a single branch, the GSB would correspond to a ~60% triplet yield for **O**, compared to 20% for **D**, a difference that cannot be easily justified. Furthermore, the triplet band is significantly broader for **O** than for **D**, contrary to what could be expected for a triplet state localized on a single branch. **Q** could not be investigated in non-polar solvents because of its insufficient solubility.

Completely different spectral dynamics take place in polar solvents, as illustrated in Fig. 4 with THF. First, a band peaking at ~2100 $cm^{-1}$ (band 1), close to the GSB, grows on a timescale similar to the fast decay component of the excitonic band. This band, observed in all polar solvents with both **O** and **Q** (Fig. 4), can thus be attributed to the symmetry-broken excited state. The allene band 2 (1915–1960 $cm^{-1}$) also rises but reaches its maximum intensity at significantly later times. In the most polar solvents, it is upshifted and its intensity is weaker. As this band develops, band 1 transforms into band 3 located at 2000–2050 $cm^{-1}$. At later times, both bands 2 and 3 decay on the hundreds of picoseconds to nanoseconds timescale, in parallel with the rise of the $T_1$ band at 2050–2060 $cm^{-1}$ (band 4). The decay of the latter is too slow to be observed within the experimental time window. These data are consistent with ES-SB followed by an alkyne–allene transformation in the branch where excitation has localized. Afterward, the allene TRICT state decays through two competitive channels: one to the alkyne ground state and one to the alkyne $T_1$ state. Band 4 is significantly narrower than in cyclohexane suggesting that, in polar solvents, the $T_1$ state is localized on a single branch.

In principle, the localization of the exciton upon ES-SB should lead to a partial refill of the GSB as one or two branches get de-excited. However, all branches in **O** and **Q** share the conjugated triazine acceptor. Therefore, ES-SB as well as the ensuing alkyne–allene transformation in one branch must affect the other branch(es), and the vibrational frequencies of their $\pi$-bridge(s) should not coincide with those of the ground state. In this case, the GSB of **O** and **Q** upon ES-SB and alkyne–allene isomerization is not expected to undergo a 2/3 or 1/2 recovery, in good agreement with the observation (Supplementary Table 9).

Based on this, bands 1 and 3 are assigned to the branch(es) of **O** and **Q**, which have lost excitation upon ES-SB and have not reacted, further on called 'spectator branches'. Band 1 is due to the spectator branches after ES-SB, whereas band 3 corresponds to these branches after alkyne–allene isomerization. Band 3 is located further from the GSB than band 1, indicating stronger perturbation. This is consistent with much larger charge-transfer character of the allene TRICT state compared to the alkyne excited state.

The intensity of band 3 relative to band 2 increases with solvent polarity. In weakly polar solvents (Supplementary Figs. 15, 16), it is only visible as a weak hump on the high-frequency side of band 2. Its intensity is similar to that of band 2 in medium-polar solvents (chloroform, THF), and dominates in highly polar media. This effect points to an increasing electron density on the spectator branches with solvent polarity. This suggests that, as solvent polarity increases, these branches participate more and more in the stabilization of the excess negative charge of the triazine core in the allene state. The number of spectator branches also explains why the relative intensity of band 3 is smaller for **O** than for **Q** (Fig. 4, Supplementary Figs. 17, 18). As only one branch participates in the delocalization of the excess charge in **Q**, a larger perturbation can be expected than for the octupolar system, where two branches are available. This leads not only to a higher intensity of band 3, but also to a larger frequency dependence on the solvent polarity ($\tilde{\nu}_{max} = 2035$–$2075$ cm$^{-1}$).

As shown in Fig. 3b, c, the alkyne–allene transformation leads to an almost complete decay of the fluorescence intensity that remains after ES-SB. The residual fluorescence spectrum consists of a weak red-shifted band, whose intensity stays almost unchanged up to 2 ns, the upper limit of the experimental time window. This shift is indicative of an emitting state with a higher charge-transfer character than the symmetry-broken alkyne state. Based on this and its lifetime, this residual emission is attributed to **O** and **Q** with one branch isomerized. By comparison, no emission could be observed after the alkyne–allene isomerization of the single-branch **D**. This difference can be accounted for by the spectator branches in **O** and **Q** that participate in the delocalization of the negative charge on the triazine core. Such emission is fully consistent with the TRIR data showing different –C≡C– stretching frequencies of the spectator branches in the symmetry-broken state and in the ensuing allene state. This is an additional indication that the spectator branches bear some excitation.

**Looking at the near-UV to near-IR region**. Further insight into the excited-state dynamics of these multipolar dyes was obtained using transient electronic absorption (TA) spectroscopy in the near-UV to near-IR region. Figure 5a, b shows evolution-associated difference spectra (EADS) obtained from a global analysis of the TA data measured with **D** and **O** in a non-polar solvent assuming a series of four successive exponential steps (A → B → C → D → ). The spectral changes resulting from the alkyne–allene isomerization of **D** are most visible in the B → C step. The band in the 600–910 nm range (EADS B, $\lambda_{max} = $ ~810 nm) originates from the triazine moiety that has a characteristic

ESA at 600–775 nm[57–59], downshifted here due to the conjugation with the $\pi$–D fragment. As this band decays, the structured band of the triazine radical anion[60] appears at 550–750 nm (EADS C, $\lambda_{max} = $ ~680 nm). In parallel, both the ESA band at ~415 nm and the stimulated emission around 465 nm (EADS B) vanish, unveiling a band in the 440–480 nm region partially masked by the GSB and due to the dialkylaniline radical cation (EADS C)[61]. Finally, the charge-separated allene state undergoes intersystem crossing to the triplet state in competition to its recombination to the alkyne ground state (C → D).

The TA spectra measured with **O** in non-polar solvents display several differences (Fig. 5b). The early spectra (EADS A and B) feature two ESA bands additionally to that of the triazine at 600–980 nm (dotted lines in Fig. 5b): a sharp one at 863 nm and another at 420–490 nm ($\lambda_{max} = $ ~450 nm). This spectrum evolves in ~40 ps into a spectrum showing the triazine radical anion at 707 nm, but still containing the 863 and 450 nm bands, although with a lower intensity (EADS C). Finally, this spectrum transforms on the ~2 ns timescale into a spectrum with a single ESA band and a residual bleach that can be associated with the T$_1$ state of **O** (EADS D). The 863 and 450 nm bands are observed in all non-polar solvents, and their time dependence is similar to those of the excitonic TRIR band and of the fluorescence (Fig. 3a). In all polar media, they can only be seen at the very earliest times and decay on the 10–100 fs timescale (Supplementary Figs. 19–24). Their presence is progressively harder to detect when going from the least polar to more polar environment. Upon their disappearance, the entire spectrotemporal evolution, as well as the signatures of the relaxed excited state, are remarkably similar for **O**, **Q** and **D** (Fig. 5d, Supplementary Fig. 21, Supplementary Table 10). This demonstrates that, upon symmetry breaking, the reaction pathway and product are the same.

Given that these 863 and 450 nm bands are not observed with the single branch **D**, and in view of their time dependence (Figs. 3a, 5b, Supplementary Fig. 24), they can be attributed to the delocalized symmetric exciton. Whereas the excitonic TRIR band is due to the S$_2$ ← S$_1$ transition, these two bands can be assigned to S$_{n>2}$ ← S$_1$ transitions. Their decay in polar solvents represents the first report of a TA signature of ES-SB in the visible region. No distinct feature could be observed in previous UV–Vis TA investigations of multipolar systems by us[30–32,34] and others[39,43,46,47]. They could be identified here not only because of the possibility to compare the TA spectra of the multipolar molecules with those of the single-branch **D** in non-polar solvents, but mostly thanks to the unambiguous spectral signatures obtained in the TRIR and FLUPS experiments. TA spectra in the UV–Vis region are often congested with ESA, GSB and stimulated emission bands, which often overlap and sometimes exhibit significant frequency shifts upon solvent relaxation. This is the reason why the NIR band at 863 nm is much more visible than the 450 nm band, which is located close to the GSB, in the region where the stimulated emission undergoes dynamic Stokes shift.

The TA data measured with **O** in non-polar solvents show that, after the initial partial decay, the delocalized excited-state bands (863 and 450 nm) follow the same dynamics as the charge-separated allene band (680 nm). This is consistent with the above-mentioned coexistence of the fluorescence with the allene IR band, and supports the establishment of an equilibrium between the symmetric excited state and the state with an isomerized branch. This contrasts with **D**, which undergoes an irreversible alkyne–allene isomerization even in non-polar solvents.

## Discussion

The combination of multiple broadband femtosecond spectroscopies reveals various stages of the photocycle of the above-

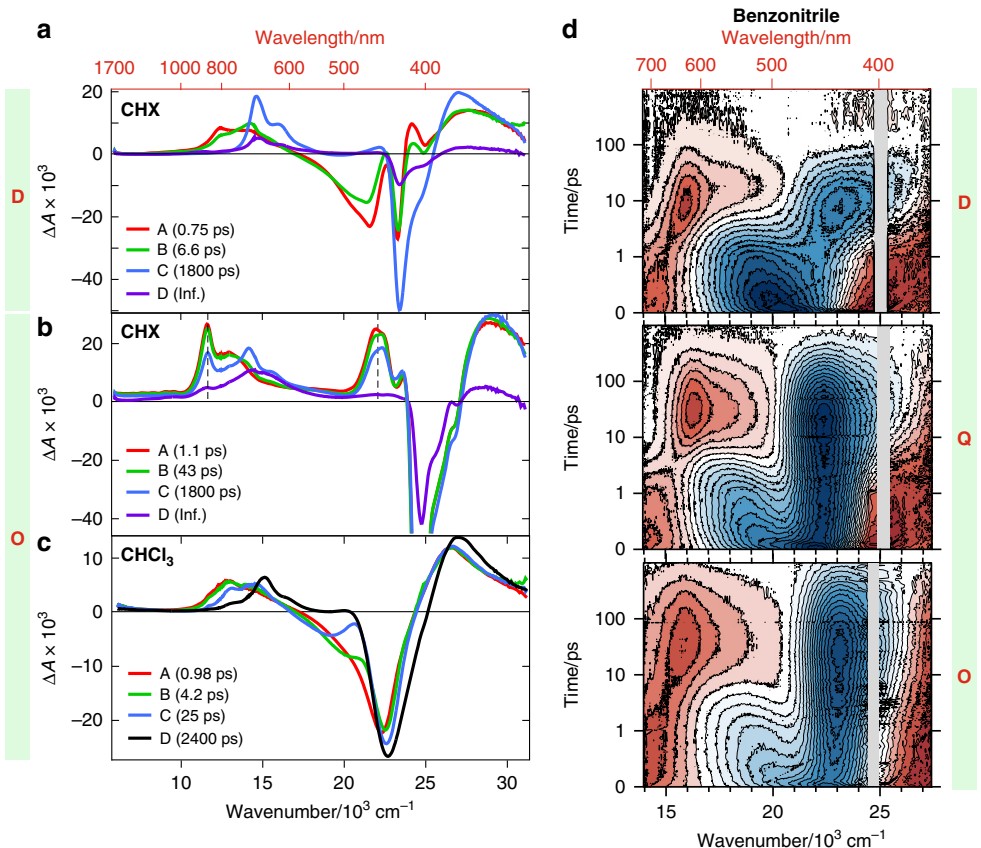

**Fig. 5 UV–Vis-NIR transient absorption spectroscopy of the multipolar compounds. a–c** Evolution-associated difference spectra obtained from a global analysis of the UV–Vis-NIR transient absorption data of **D** and **O** in non-polar (**a**, **b**) and of **O** in polar (**c**) media for a consecutive A → B → C → D → scheme. In non-polar media, the evolution-associated difference spectra (EADS) D is assigned to the triplet state, whereas in polar solvents it is attributed to the relaxed allene product. **d** Transient absorption contour plots of **D**, **Q**, **O** in the highly polar benzonitrile. Positive excited-state absorption bands are in red, whereas ground-state bleach and stimulated emission are in blue.

mentioned multipolar molecules in great detail. Both the ground and Franck–Condon excited states of **O** and **Q** are symmetric and multipolar, as shown by the one- and two-photon absorption spectra. The delocalized symmetric multipolar exciton is fundamentally unstable in polar environment due to the asymmetry and strength of the fluctuating solvent field. It collapses into a localized dipolar exciton that is efficiently stabilized by solvation, independently of its original multipolar nature. The whole ES-SB process is, thus, driven by solvation dynamics. This is clearly shown here with both octupolar and quadrupolar dyes, which exhibit the same ES-SB dynamics in each particular polar solvent. This suggests that ES-SB should occur similarly in larger or higher-dimensional weakly-coupled systems, such as conjugated macrocyclic rings and ribbons, and multichromophoric arrays.

The $S_2 \leftarrow S_1$ excitonic electronic transition is an unequivocal signature of the delocalized exciton and gives a direct and unambiguous access to ES-SB. Its energy reflects the Davydov splitting and, depending on the interbranch coupling, it can be found in the mid- to near-IR region. The previous electronic TA investigations have focused on the much higher-energy UV–Vis range, where the spectra are usually congested with local transitions that do not report on ES-SB. Here however, $S_{n>2} \leftarrow S_1$ transitions between delocalized excited states could be identified in the near-IR and visible regions. They are also sensitive to symmetry breaking as they persist during the whole excited-state lifetime in non-polar solvents, but vanish very rapidly in polar media. Observation of such high-energy signatures of delocalization is rather exceptional, contrary to the mid-IR $S_2 \leftarrow S_1$

excitonic transition. The latter is a direct consequence of the coupling, and is, thus, a more general feature. Since the electronic coupling is experimentally accessible via the comparison of OPA and TPA spectra, the energy of this transition can be predicted and its dynamics can be used to monitor localization precisely. This is clearly not the case of the higher-energy excitonic bands whose positions cannot be easily predicted. ES-SB in polar solvents is also visible in the FLUPS experiments as a decrease of the $S_1 \rightarrow S_0$ emission transition dipole.

The $S_2 \leftarrow S_1$ mid-IR absorption observed here bears some resemblance with the low-energy polaron bands observed in the electronic absorption spectra of open-shell cations of conjugated oligomers and in doped conjugated polymers, as the latter report on the delocalization of the charge across the chain[62,63]. Charge localization breaks the symmetry leading to the transformation of Raman active modes into infrared-active vibrations (IRAVs)[64–66].

Both TRIR and UV–Vis-NIR experiments reveal the occurrence of the photochemical charge-separating alkyne–allene reaction with the multipolar molecules, provided solvent-induced SB has taken place beforehand. In non-polar solvents, where SB is not observed, the allene product is in equilibrium with the delocalized excited state. In contrast, the alkyne–allene isomerization is irreversible with the single-branch **D**. The different photoreactivity of **D** and **O** arises from the presence of additional branches in the latter. The alkyne–allene isomerization was shown to be operative with **D** because of the high charge-transfer character of the excited state. The charge-transfer character of a single branch of **O** in the delocalized coherent exciton is

obviously smaller than in a localized dipolar state. As a consequence, a higher barrier for isomerization is expected for **O** than **D**. This is fully supported by the rise time of the allene IR band of **O** that is about 10 times slower than for **D** ($\tau_r^{eff} = 40$ vs. 4 ps in cyclohexane; $\tau_r^{eff} = 92$ vs. 9 ps in hexadecane). As isomerization involves large amplitude twisting motion, $\tau_r^{eff}$ increases with solvent viscosity, as discussed in detail in the Supplementary Discussion.

The relative amplitude of the fast decay component of the delocalized exciton population points to an equilibrium constant between this state and the allene state of 0.59. Therefore, isomerization is 70% slower than the back reaction, allowing an estimation of the isomerization time constant of about $2.7\tau_r^{eff} = 110$ ps in cyclohexane and 250 ps in hexadecane. This is about 30 times slower than for **D**. Based on this, the activation barrier for the isomerization in **O** should be larger than that for **D** by about 700 cm$^{-1}$ (3.4 $k_B T$ at room temperature). Furthermore, given an equilibrium constant of 0.59, the allene state of **O** should be only about 110 cm$^{-1}$ (0.53 $k_B T$ at room temperature) above the delocalized excited state. Finally, according to the Kasha

excitonic model (Fig. 1c), the delocalized excited state should be more stable than the localized excited state by $V = 880$ cm$^{-1}$. All these numbers can be used to sketch the energy-level scheme depicted in Fig. 6, which is fully consistent with the results obtained for both **O** and **D** in non-polar solvents. This scheme suggests that the transition state for the alkyne–allene isomerization is rather similar for both molecules. It is most probably associated with a large charge-transfer character. Access to this state should be facile from the dipolar alkyne excited state of **D**, but not from the delocalized excited state of **O** where the charge-transfer character in each branch is markedly smaller.

This scheme also accounts for the equilibrium observed with **O**. Whereas the energy of the allene state can be expected to not differ much for **D** and **O**, that of the delocalized alkyne excited state of **O** is stabilized relative to **D** by the excitonic coupling $V$. Because this stabilization energy is suppressed upon alkyne–allene isomerization of **O**, this process is not very efficient in non-polar solvents. The energy gained upon formation of the allene product of **O** is not sufficient to fully compensate for the loss of excitonic interaction energy, and the process is endergonic.

All previous investigations revealed that ES-SB was not occurring in non-polar solvents[30–36]. Symmetry breaking takes place in non-polar solvents for these reactive systems, although not very efficiently, only because the chemical transformation stabilizes the symmetry-broken allene state. This case is exceptional, because in all systems investigated so far, the stabilization of the symmetry-broken state was due to solvation. Without solvation energy, the delocalized excited state was more stable than the symmetry-broken state, because of the excitonic coupling. In a Jahn–Teller picture of ES-SB, exciton localization is driven by the energy stabilization gained upon structural distortion. For this to be operative, this energy gain should exceed the excitonic coupling energy that is lost upon localization[67]. This is not the case for both **O** and **Q** in the absence of alkyne–allene isomerization, as well as in the previously studied quadupolar molecules. In these cases, exciton localization is driven by the gain in solvation energy, which leads to a strong stabilization of the dipolar exciton. The fact that ES-SB is observed with **O** in non-polar solvents is due to the alkyne–allene isomerization. This substantial structural distortion allows for a significantly larger energy stabilization in non-polar environments that almost compensates for the loss of excitonic interaction energy.

Therefore, the ES-SB process taking place here with **O** in non-polar solvents is chemically-driven instead of solvent-driven. This bespeaks the fragility of the coherent delocalized exciton even in

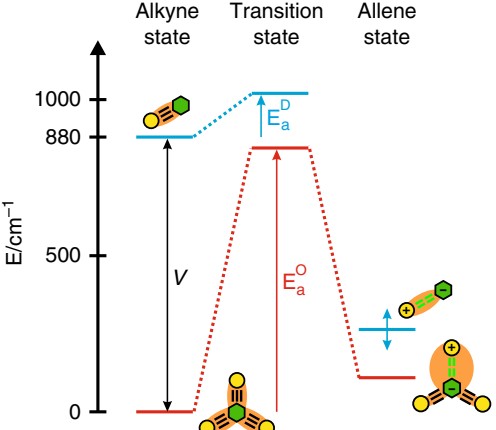

**Fig. 6 Tentative energy-level scheme for the isomerization of D and O in non-polar solvents.** Green hexagon represents the triazine electron acceptor, yellow circles are the electron donors and orange shading is the electronic excitation. The allene state is a charge-separated state. Only the difference between the activation energies $E_a^D - E_a^O$ can be estimated, but their absolute values are unknown, as is the relative energy of the allene state of **D**.

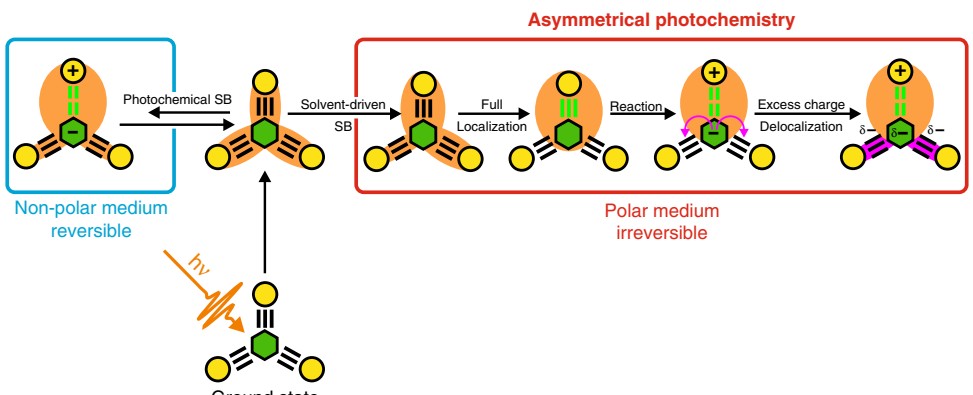

**Fig. 7 Schematic representation of the excited-state processes in the octupolar dye O.** Green hexagon represents the triazine electron acceptor, yellow circles are the electron donors and the orange shading stands for the electronic excitation. Photochemical symmetry breaking is reversible whereas solvent-driven SB is irreversible and leads to the efficient funneling of the excitation into a single arm where it drives chemical reaction (charge separation coupled with the alkyne–allene isomerization). The excess negative charge acquired by the triazine unit as a result of the reaction is delocalized back into the spectator arms (magenta arrows and shading).

the absence of high fluctuating electric fields. As a consequence, the symmetry-broken incoherent dipolar state is chemically distinct from the symmetric coherent multipolar excited state, contrary to the previous cases (Fig. 7).

The situation changes considerably when going to polar solvents. Because of solvation, the symmetry-broken alkyne state of **O** and **Q** is more stable than the delocalized state. Consequently, solvent-driven SB occurs in all polar solvents. The localized excited alkyne state can be clearly observed spectroscopically, contrary to the non-polar solvents (Fig. 7).

As the charge-transfer character increases continuously on the way to the TRICT isomerization product, the activation energy decreases with increasing solvent polarity, whereas the driving force increases. Consequently, the reaction becomes faster and irreversible. Solvent-driven SB, which results in a highly polar localized excited state, is therefore a prerequisite for the reaction to be efficient in the multipolar molecules.

The TRIR and FLUPS results reveal that the nature of the allene product state of **D** and of the multibranched molecules differs. Whereas the reactive branch of **O** and **Q** undergoes isomerization, their spectator branches participate in the stabilization of the negative charge, which is confined on the triazine subunit in **D**. As a result, the allene product state of **D** is non-emissive, whereas weak but distinct emission can be observed with the allene products of **O** and **Q**.

This charge delocalization has a significant impact on the lifetime of the charge-separated allene photoproduct. In the case of **D**, the lifetime of this state varies from a few tens of picoseconds in highly polar solvents up to few hundreds ps in the lowest polar solvents. Its lifetime increases by a factor 7–10 for **Q** and >15 for **O** (except in the highest polarity solvents), reaching a value of several nanoseconds (Supplementary Table 10). This illustrates the importance of the stabilization of the charge-separated product in the aftermath of symmetry breaking and ensuing reaction. The decay of the charge-separated allene state can be viewed as a charge recombination process. In terms of Marcus electron transfer theory, the dilution of the negative charge upon its delocalization on the spectator branch(es) leads to a decrease of the electronic coupling for charge recombination, hence a slower recombination. An allene photoproduct with a lifetime of several nanoseconds is an important milestone, because it is long-lived enough to allow diffusion-limited reactions to take place. Therefore, this photochemical reaction of the multipolar molecules could be applied as a source of in-situ generated allene-type intermediates for multistep preparative one-pot organic reactions.

In a broad context, our findings have significant implications for the design and synthesis of advanced functional systems and the control of photochemical transformations. They directly evidence functional asymmetry stemming from the nature of the relaxed exciton. They raise the question of whether coherence in the excited state can be functionally relevant or whether it is too fragile to be utilized in realistic photochemical transformations. The local asymmetry arising from local fields is at the core of the functional asymmetry of natural photosynthetic reaction centers[68]. It also explains the lack of success in the elaboration of symmetric molecular systems that could exhibit double excited-state proton transfer[69–71]. Solvent-driven SB leads to a concentration of the excitation on a particular area of a symmetric molecule and results in a strong enhancement of its photochemical reactivity providing effective means of guiding the latter in addition to mode-selective IR excitation[72–75]. Without symmetry breaking, photochemistry is significantly less efficient, if operative at all, because the excitation density, being evenly distributed over the molecule, is not sufficient to drive a chemical reaction. The concentration of the excitation in a limited area of

the molecule, which allows for highly efficient and fast asymmetrical photochemistry, is the main practical outcome of ES-SB. Therefore, controlling ES-SB in multichromophoric systems opens the possibility to direct the electronic excitation into a specific position, just where the photochemical reaction is wanted.

## Methods

**Samples**. The synthesis of **D**, **Q** and **O**, the solvent characteristics and additional technical details are described in the Supplementary Information.

**Two-photon excited fluorescence (TPEF) spectroscopy**. Excitation was performed using the output of a collinear OPA (TOPAS-Prime, Light Conversion) equipped with a NirUVis mixing unit and pumped at 800 nm (100 fs, 1kHz) with a Ti:Sapphire amplifier system (Spitfire, Spectra-Physics). Two-photon excited fluorescence was collected at 90° by a broadband dielectric concave mirror, focused onto the entrance of a grating spectrograph (Newport) and detected by a multipixel silicon avalanche photodiode (400 pixels, Hamamatsu). The residual excitation beam was directed to a calibrated powermeter equipped with a thermal sensor (Thorlabs) to monitor beam intensity on a shot-to-shot basis. Excitation pulse energies were of the order of few microjoules. The quadratic dependence of the fluorescence intensity on the excitation irradiance was routinely checked at few selected wavelengths.

**Time-resolved infrared (TRIR) spectroscopy**. The setup used to record the femtosecond TRIR spectra was described in Dereka et al.[30] It consisted of a 1 kHz Ti:Sapphire amplifier (Spectra Physics Solstice) generating 100 fs pulses at 800 nm. Excitation was carried out at 400 nm with pulses generated upon frequency doubling part of the amplifier output. The polarization of the pump pulses was adjusted at magic angle relative to the IR probe pulses using a combination of Glan–Taylor polarizer and zero-order half-wave plate. These elements limited the time resolution of the experiment to about 300 fs. The pulses were focused onto the sample to a 350 μm spot, corresponding to an irradiance of 0.2–0.6 mJ cm$^{-2}$. The amplitude of the signal was checked to scale linearly with the excitation irradiance. The mid-IR probe pulses were produced by difference frequency mixing of the output of an optical parametric amplifier (Light Conversion, TOPAS-C with NDFG module) pumped at 800 nm. A wire-grid polarizer was used to control their polarization. Reflections on a CaF$_2$ wedge produced two horizontally polarized IR beams that were focused onto the sample to a 140 μm diameter spot. One of them was overlapped with the pumped region of the sample, whereas the other was used as a reference. Both of them were then focused onto the entrance slit of an imaging spectrograph (Horiba, Triax 190, 150 lines per mm) equipped with a liquid nitrogen cooled 2×64 element MCT array (Infrared Systems Development), allowing for a ~3 cm$^{-1}$ resolution. Each data points corresponded to the average of 2000 signal shots. This procedure was applied five to ten times. A flow cell was used to refresh the sample solutions after each shot. The absorbance at 400 nm on a 200 μm optical pathlength was between 0.2 and 0.8. Given the bandwidth of the IR pulses (~400 nm), transient spectra over the 1650–2850 cm$^{-1}$ spectral region were obtained by merging up to 9 spectra recorded within a ca. 150–200 cm$^{-1}$ spectral window. To detect possible problems during the measurements, the datasets were compared before being averaged and merged to obtain a single spectrum over a broad spectral window. No tail matching was required. The samples did not exhibit any degradation throughout the experiment.

**UV–vis transient absorption (UV–Vis TA)**. Excitation was performed using 400 nm pulses generated by frequency doubling part of the output of a 1 kHz Ti: Sapphire amplified system (Spectra Physics, Solstice Ace). The polarization of the pump pulses was set to magic angle relative to the white-light pulses. Probing was achieved using white-light pulses generated by focusing the 800 nm pulses in a CaF$_2$ plate. All transient absorption spectra were corrected for background signals showing up before time zero and for the dispersion due to the optical chirp. The samples were hold in 1 mm quartz cuvettes and bubbled with nitrogen gas during the measurements giving a wavelength-dependent IRF of ~80–350 fs. The absorbance of the sample at the excitation wavelength was 0.2–0.5 (at 400 nm) on 1 mm. The absorption spectra of all samples before and after the measurements showed no evidence of degradation.

**Near-infrared (NIR) transient absorption (NIR TA)**. The setup was based on the same scheme as the UV–Vis TA, the same laser source and identical experimental procedures were applied. The probe white light was generated by focusing 800 nm pulses in a YAG crystal and was then split into a reference and a sample beam by a reflective metallic neutral density filter. After passing the sample, the beam was dispersed in a home-built prism spectrometer and the intensity recorded with an InGaAs detector. For merging the spectra recorded with the UV–Vis and NIR TA setups, the signal in the overlap region between 690 and 740 nm was compared and one of the two datasets was multiplied with a constant factor accounting for the

difference in pump power. The comparison of the kinetics recorded in the overlap region additionally serves as quality control.

**Fluorescence up-conversion spectroscopy (FLUPS)**. Excitation was performed with 100 fs pulses at 400 nm generated by frequency doubling part of the output of a standard 1 kHz Ti:Sapphire amplified system. The pump intensity on the sample was below 1 mJ cm$^{-2}$. The gate pulses were at 1340 nm and were produced by an optical parametric amplifier (TOPAS-C, Light Conversion). Detection of the up-converted spectra was performed with a CCD camera (Andor, DV420ABU). The FWHM of the cross correlation of the gate with the solvent Raman signal was ~170 fs. Corrected time-resolved emission spectra were obtained by calibration with secondary emissive standards. The temporal chirp was determined measuring the instantaneous response of BBOT (Radiant Dyes) in all the solvents used.

## Data availability
The data shown in the figures can be downloaded from https://doi.org/10.5281/zenodo.3567554.

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

## Acknowledgements
This research was supported by the Swiss National Science Foundation, Grant Number 200020–184607, and the University of Geneva. B.D. acknowledges the support from the Swiss National Science Foundation through Postdoc.Mobility fellowship Grant P400P2_180765. D.S. and A.M.M. acknowledge support from the Nebraska Research Initiative. M.L. acknowledges the support from the TU Wien "Innovative Projects" research funds.

## Author contributions
B.D. conceived the idea and designed experiments. D.S. synthesized methyl-substituted **D**, **Q** and **O**. M.L. synthesized butyl-substituted **D** and **O**. B.D. performed all steady-state, TCSPC and TRIR measurements. B.D., A.R. and A.A. performed UV–Vis TA measurements. A.A. performed NIR TA measurements. A.R. performed FLUPS measurements. B.D. and A.R. performed TPA measurements. B.D. analyzed the data. R.L. and A.M.M. supervised organic syntheses. E.V. supervised and guided the whole project. All authors discussed the results, analyses and interpretations. B.D. and E.V. wrote the paper with input from all authors.

## Competing interests
The authors declare no competing interests.
