## [Peer Review File · Nature Communications]

Reviewers' comments:

Reviewer #1 (Remarks to the Author):

The work presented is an exemplary study of a fundamental problem of symmetry breaking in excited states. It is fundamentally interesting due to many hypothesis about such mechanisms being proposed, from a more "classical" Robin-Day classifications in mixed-valence compounds, to coherence in photosynthesis. The work by Vauthey et al describes an elegant application of ultrafast TRIR spectroscopy to resolve symmetry-breaking in non-quadrupole molecules. The principal novelty lies precisely in observation, and the proof of, symmetry-breaking in the excited state of tri-alkene systems, which depending on the solvent polarity, can lead to different outcomes. The data are obtained and analysed most thoroughly. Multiple ultrafast methods have been used to support the conclusions. The results demonstrate that solvent-driven symmetry-breaking is another route to control photophysical behaviour and photoreactivity. The work is strongly suggested for publication in Nature Communications.

The suggestion for the authors is to discuss coherence/decoherence in the main text. This central concept to their work and to its significance appears in the introduction, and then in references. It needs to linked directly to the findings. How viscosity of the solvent has been taken into account (important for rotation) needs to be mentioned.

Otherwise, the work can be published as is.

Reviewer #2 (Remarks to the Author):

This paper, entitled "Solvent Tuning of Photochemistry upon Excited-State Symmetry Breaking", reports the modulation of photo-reactivity of large conjugated molecules by controlling the localization of excitation through the excited-state symmetry breaking (ES-SB). First, OPA and TPA measurements ensure that these multipolar compounds are symmetric in the ground and FC excited state. Next, the authors have investigated the ES-SB processes using TR-IR, FLUPS, and fs-TA. Interestingly, in nonpolar solvents, the equilibrium between delocalized S1 and allene states was observed for Octapolar compound. Based on this, the authors suggest that the ES-SB in nonpolar solvents is

chemically-driven instead of solvent-driven because the chemical transformation stabilizes the SB-allene state in nonpolar medium. On the other hand, polar medium can stabilize the SB states more significantly than that in nonpolar medium because of the solvation. Finally, the authors concluded

that “this photochemical reaction of the multipolar molecules could be applied as a new source of in situ generated allene-type intermediates for multistep preparative one-pot organic reactions”.

Eric Vauthey et al. have published a plenty of similar works on ES-SB using

electronic and/or vibrational spectroscopies. However, the concept of this work is highly novel, and this manuscript was very well written. This paper seems to be the essence of ES-SB.

Especially, the flow of this manuscript is very clear. Therefore, I strongly suggest that this work is suitable for publication in ‘Nature Communications’ after minor revisions. However, several comments are mentioned below which the authors should address for further revision.

1) In Figure 1b, the absorption spectra were measured in CHX? I found different absorption spectra in Figure S1 compared to Figure 1b. In Figure S1, the distinct vibronic features are obvious, but the absorption lineshapes in Figure 1b should be different.

2) In SI, the authors explained large solvatochromism of O and Q in the absorption spectra.

And a lot of data suggest that O and Q are not followed by point dipole/multipole approximation.

The authors used Ivanov symmetry-breaking model to describe the possibility of ES-SB of O and Q in weakly polar solvents. To the best of my knowledge, Ivanov model also assumes the point dipole approximation. Then, is the Ivanov model suitable for your compounds? (In related references, all compounds seem to follow point dipole approximation.)

3) In line with comment 2, the large solvatochromism and the breakdown of point-dipole approximation are unusual based on your previous researches. I wonder what factors make the differences. In my opinion, the relative sizes of the solute and solvents are not the main factors because other quadrupolar compounds (either D and/or A) are also much larger than the solvent.

4) The authors assume that the symmetry is preserved in the ground and FC excited states. However,

the ground state absorption spectra in polar mediums show large solvatochromism. These results may suggest that the bond order is perturbed by solute-solvent interactions, which can break the ground state symmetry. Did the authors try to measure TPA in polar medium? The solvent dependent TPA data should strengthen the authors' suggestion.

5) In Figure 3c, the authors suggest that the relative fluorescence intensities of O and Q between delocalized and TRICT states indicate the localization. I'm wondering how the direct comparison between population and maximum intensity obtained from the global analysis could explain the author's assignment. Furthermore, O and Q show residual fluorescence from the higher CT states. In this regard, the exact values of 1/3 or 1/2 for O and Q, respectively, are not so convincing.

6) In Figure S27, the spectral evolution was observed before zero time. I think that the authors need chirp- correction or zero-correction for clarity.

Reviewer #3 (Remarks to the Author):

This work shows for the first time experimental evidence of symmetry breaking driven by a direct photochemical reaction. The authors have utilised an impressive area of state of art spectroscopic techniques to clearly demonstrate this fact. The system investigated here is important in its own right but generally serves as a proxy for donor-pi-acceptor multichromophoric arrays. These results are important in the area of conjugated organic materials for optoelectronic and photovoltaics. This work is of potential importance to designing photoactive materials and highlights the types of molecular functionalities that should be studied as sources of switching the symmetry breaking on or off. The work has been well performed and the paper is well written and easy to follow for the reader with general knowledge of photochemistry and photophysics. I believe that this work significantly advances what has been observed for such conjugated systems, and sets new

benchmarks in the level of detail that should be sought from the molecular photochemistry of complex systems.

I have a couple of minor points that I would like the authors to consider:

Could they have some discussion on the localised and delocalised excitonic picture used in the paper by mapping this to a molecular excited state where one would normally consider symmetry breaking features as occurring through vibronically coupled (quasi-Jahn-Teller type) interactions between degenerate electronic states at high-symmetry. Is it correct to think of degenerate excitons (or some linear combination thereof) causing a symmetry breaking in complete analogy to Jahn-Teller type molecular case? If this is the case what are the molecular analogues of vibronic coupling leading to no-symmetry breaking? Is this just a highly dynamical system pseudo-rotating freely around a conical intersection very close in energy.

Reviewer #1 (Remarks to the Author):

The work presented is an exemplary study of a fundamental problem of symmetry breaking in excited states. It is fundamentally interesting due to many hypothesis about such mechanisms being proposed, from a more "classical" Robin-Day classifications in mixed-valence compounds, to coherence in photosynthesis. The work by Vauthey et al describes an elegant application of ultrafast TRIR spectroscopy to resolve symmetry-breaking in non-quadrupole molecules. The principal novelty lies precisely in observation, and the proof of, symmetry-breaking in the excited state of tri-alkene systems, which depending on the solvent polarity, can lead to different outcomes. The data are obtained and analysed most thoroughly. Multiple ultrafast methods have been used to support the conclusions. The results demonstrate that solvent-driven symmetry-breaking is another route to control photophysical behaviour and photoreactivity. The work is strongly suggested for publication in Nature Communications.

Authors answer: we are very pleased that the referee appreciated this work.

The suggestion for the authors is to discuss coherence/decoherence in the main text. This central concept to their work and to its significance appears in the introduction, and then in references. It needs to linked directly to the findings.

Authors answer: We have modified the text accordingly by inserting several sentences at the appropriate places throughout the text:

p.4: 'The strong solvatochromism points to a dipolar excited state, whereas the decrease of transition dipole moment is consistent with a localization of the **excitation, i.e. with the decoherence of the exciton.**³⁶

p.5: 'The disappearance of this $S_2 \leftarrow S_1$ band reflects the loss of interbranch coupling as a consequence of the localization of the excitation on a single branch. **This is a direct spectroscopic manifestation of the decoherence of the initially prepared exciton.**'

p.13: 'The charge-transfer character of a single branch of O in the delocalized **coherent exciton** is obviously smaller than in a localized dipolar state.'

p.16: 'Therefore, the ES-SB process taking place here with O in non-polar solvents is chemically-driven instead of solvent-driven. **This bespeaks the fragility of the coherent delocalized exciton even in the absence of high fluctuating electric fields.** As a consequence, the symmetry-broken **incoherent dipolar** state is chemically distinct from the symmetric **coherent multipolar** excited state, contrary to the previous cases (Figure 7).'

p.18: 'In a broad context, our findings have significant implications for the design and synthesis of advanced functional systems and the control of photochemical transformations. They directly evidence functional asymmetry stemming from the nature of the relaxed exciton. **They raise the question of whether coherence in the excited state can be functionally relevant or whether it is too fragile to be utilized in realistic photochemical transformations.**'

How viscosity of the solvent has been taken into account (important for rotation) needs to be mentioned.

Authors answer: The effect of solvent viscosity is discussed in detail in the supporting information (Section S3.1.2). We indeed do recognize the importance of solvent friction on the alkyne-allene isomerization timescale as it involves large-amplitude twisting motion. Section S3.1.2 of the supplementary information discusses the viscosity effect on the isomerization time constant for the dipolar dyad **D**. Figures S8, S10 and S11 illustrate the effect of viscosity in polar and non-polar media according to eq. S4. The effect of viscosity is similar for the multipolar systems in polar media. However, we do not discuss it explicitly because it is not always possible to separate the symmetry-breaking and reaction timescales as they often overlap (as indicated in Figs. 4, S15 and S16 by overlapping colorbars representing different processes). In addition, as discussed in section S3.1.2, the polarity of the environment modulates the reaction rate (Fig. S8) because isomerization is accompanied by intramolecular charge separation. Therefore, it would only be possible to cleanly correlate solvent viscosity with the reaction rate for **Q** and **O** in non-polar solvents (as in Fig. S10). We have done experiments only in two non-polar solvents: cyclohexane and hexadecane and we observe that the effective isomerization time constant for **O** is about 30 times slower in both these solvents compared to **D** (as stated in the main text on page 15). We don't have TRIR data for many more apolar solvents to show this quantitative correlation explicitly.

To stress the effect of viscosity in the main text, the following sentence was added (p.15):

'This is fully supported by the rise time of the allene IR band of **O** that is about 10 times slower than for **D** ($\tau^{eff} = 40$ vs. 4 ps in cyclohexane; $\tau^{eff} = 92$ vs. 9 ps in hexadecane). **As isomerization involves large amplitude twisting motion, τ^{eff} increases with solvent viscosity, as discussed in detail in Section S3.1.2.'**

Reviewer #2 (Remarks to the Author):

This paper, entitled "Solvent Tuning of Photochemistry upon Excited-State Symmetry Breaking", reports the modulation of photo-reactivity of large conjugated molecules by controlling the localization of excitation through the excited-state symmetry breaking (ES-SB). First, OPA and TPA measurements ensure that these multipolar compounds are symmetric in the ground and FC

excited state. Next, the authors have investigated the ES-SB processes using TR-IR, FLUPS, and fs-TA.

Interestingly, in nonpolar solvents, the equilibrium between delocalized S1 and allene states was observed for Octapolar compound. Based on this, the authors suggest that the ES-SB in nonpolar solvents is

chemically-driven instead of solvent-driven because the chemical transformation stabilizes the SB- allene state in nonpolar medium. On the other hand, polar medium can stabilize the SB states more significantly than that in nonpolar medium because of the solvation. Finally, the authors concluded that "this photochemical reaction of the multipolar molecules could be applied as a new source of in situ generated allene-type intermediates for multistep preparative one-pot organic reactions".

Eric Vauthey et al. have published a plenty of similar works on ES-SB using electronic and/or vibrational spectroscopies. However, the concept of this work is highly novel, and this manuscript was very well written. This paper seems to be the essence of ES-SB.

Especially, the flow of this manuscript is very clear. Therefore, I strongly suggest that this work is suitable for publication in 'Nature Communications' after minor revisions.

Authors answer: We thank the reviewer for his very positive assessment of our work.

However, several comments are mentioned below which the authors should address for further revision.

1) In Figure 1b, the absorption spectra were measured in CHX? I found different absorption spectra in Figure S1 compared to Figure 1b. In Figure S1, the distinct vibronic features are obvious, but the absorption lineshapes in Figure 1b should be different.

Authors answer: This is indeed a mistake. The spectra shown in Figure 1b were taken in chloroform and not in cyclohexane. The one-photon absorption spectra in cyclohexane are shown in Figure S1. We thank the reviewer for his perspicacity.

2) In SI, the authors explained large solvatochromism of O and Q in the absorption spectra.

And a lot of data suggest that O and Q are not followed by point dipole/multipole approximation.

The authors used Ivanov symmetry-breaking model to describe the possibility of ES-SB of O and Q in weakly polar solvents. To the best of my knowledge, Ivanov model also assumes the point dipole approximation. Then, is the Ivanov model suitable for your compounds? (In related references, all compounds seem to follow point dipole approximation.)

Authors answer: When we wrote that these large molecules cannot be described as large point multipoles, we were thinking about **O** and **Q** that cannot be described as point octupole and point quadrupole, respectively. Our data do not allow making any conclusion on the point-dipole approximation for **D**.

To avoid any confusion, the sentence in the main text has been rewritten:

'This solvatochromism can be explained by considering that **O and Q cannot be described as point octupole or point quadrupole, respectively** (Supplementary Section 2.1).'

In any case, the Ivanov model uses the point -dipole approximation as a simplification to avoid explicit consideration of solvent molecules size and shape. In principle, it does not matter because even if the point-dipole approximation was not valid it would not change the model prediction since all that matters is the energy difference between symmetric and symmetry-broken states due to solvation. Even if the point-dipole model was poor, it should similarly affect both energies.

3) In line with comment 2, the large solvatochromism and the breakdown of point-dipole approximation are unusual based on your previous researches. I wonder what factors make the differences. In my opinion, the relative sizes of the solute and solvents are not the main factors because other quadrupolar compounds (either D and/or A) are also much larger than the solvent.

Authors answer:

This is a good point. The major difference between the molecules investigated here and those studied previously is the larger dipole moment of the D- π -A branches in both the ground and the excited states. This is due to the triazine in **O** and the chlorotriazine in **Q**, that are stronger acceptors than cyanobenzene and fluorenone in the previously studied

quadrupolar molecules. In fact, the alkyne-allene reaction is precisely due to the very large charge-transfer character in the excited aniline- π -triazine branch. This isomerization is not observed for example in the fluorenone-based quadrupolar system because of the weaker charge-transfer character of the excited aniline- π -fluorenone branches. Therefore, both the size of the molecules and the presence of a strong electron acceptor are at the origin of the observed breakdown of the point-quadrupole/octupole approximations. This strong dipolar character of the individual branches of **O** and **Q** is mentioned in the SI (p. S16).

4) The authors assume that the symmetry is preserved in the ground and FC excited states. However, the ground state absorption spectra in polar mediums show large solvatochromism. These results may suggest that the bond order is perturbed by solute-solvent interactions, which can break the ground state symmetry. Did the authors try to measure TPA in polar medium? The solvent dependent TPA data should strengthen the authors' suggestion.

Authors answer: As discussed above, the TPA spectra are in chloroform and not in cyclohexane. Therefore, the comparison between the TPA and OPA spectra suggests that both **O** and **Q** have symmetric and multipolar ground state and Franck-Condon excited states in a medium polar solvent.

The ensemble of time-resolved data (especially the TRIR and FLUPS) show unambiguously that that photo-excitation initially populates a symmetric and multipolar excited state.

We can safely exclude bond-order perturbation by solute-solvent interactions. The stationary IR spectra in the C \equiv C stretching region (Fig. S5) shows that the alkyne character in **D** is preserved in different solvents. The C \equiv C stretching band only manifests some solvatochromism due to the variation of the local field (Stark effect).

5) In Figure 3c, the authors suggest that the relative fluorescence intensities of O and Q between delocalized and TRICT states indicate the localization. I'm wondering how the direct comparison between population and maximum intensity obtained from the global analysis could explain the author's assignment. Furthermore, O and Q show residual fluorescence from the higher CT states. In this regard, the exact values of 1/3 or 1/2 for O and Q, respectively, are not so convincing.

Authors answer: Localization can be deduced by comparing the spectral amplitude of the delocalized and localized states (red and blue spectra in Figure 3c) but not by comparing the spectra of the delocalized and TRICT states (red and black in Figure 3c). This is explicitly discussed in ref. 36. This argument relies on the assumption that the timescales of symmetry breaking and isomerization are well separated (because the reaction causes an almost complete quenching of the fluorescence). This is the case in benzonitrile shown in Figure 3c and the evolution-associated spectra obtained from global analysis can be used as proxies of the species-associated spectra. Fig. S26 shows that this assumption works also rather well in chloroform. In the case of benzene, also shown in Fig. S26, the distinction is not as clear because symmetry breaking and isomerization have partially overlapping timescales.

Higher CT states do not contribute here because they are not populated. We excite the S₁←S₀ transition that is well separated from higher-energy transitions and the residual fluorescence is exclusively from the weakly emissive TRICT state.

6) In Figure S27, the spectral evolution was observed before zero time. I think that the authors need chirp- correction or zero-correction for clarity.

Authors answer: We think that the reviewer refers to Figure S24. Our data are all chirp-corrected and time-zero is set properly as described in Section S1.3 of the SI. However, we are limited by the pulse duration and have a finite time resolution. Additionally, the instrument response function in the NIR spectral region is worsened by the group-velocity mismatch between the 400 nm pump and NIR probe. What Fig. S24 shows is the spectral evolution that happens within the instrument response. Of course, we cannot rigorously analyze it quantitatively and we don't do that. However, we can discuss it qualitatively to support an argument that is in any case evident from the data shown in apolar solvents, where this spectral feature persists for long time.

Reviewer #3 (Remarks to the Author):

This work shows for the first time experimental evidence of symmetry breaking driven by a direct photochemical reaction. The authors have utilised an impressive area of state of art spectroscopic techniques to clearly demonstrate this fact. The system investigated here is important in its own right but generally serves as a proxy for donor-pi-acceptor multichromophoric arrays. These results are important in the area of conjugated organic materials for optoelectronic and photovoltaics. This work is of potential importance to designing photoactive materials and highlights the types of molecular functionalities that should be studied as sources of switching the symmetry breaking on or off. The work has been well performed and the paper is well written and easy to follow for the reader with general knowledge of photochemistry and photophysics. I believe that this work significantly advances what has been observed for such conjugated systems, and sets new benchmarks in the level of detail that should be sought from the molecular photochemistry of complex systems.

Authors answer: we wish to thank the reviewer for the high appraisal of this work.

I have a couple of minor points that I would like the authors to consider:

Could they have some discussion on the localised and delocalised excitonic picture used in the paper by mapping this to a molecular excited state where one would normally consider symmetry breaking features as occurring through vibronically coupled (quasi-Jahn-Teller type) interactions between degenerate electronic states at high-symmetry. Is it correct to think of degenerate excitons (or some linear combination thereof) causing a symmetry breaking in complete analogy to Jahn-Teller type molecular case? If this is the case what are the molecular analogues of vibronic coupling leading to no-symmetry breaking? Is this just a highly dynamical system pseudo-rotating freely around a conical intersection very close in energy.

Authors answer: we added the following paragraph in the main text (p.16)

***In a Jahn-Teller picture of ES-SB, exciton localization is driven by the energy stabilization gained upon structural distortion. For this to be operative, this energy gain should exceed the excitonic coupling energy that is lost upon localization.⁶⁶ This is not the case in both **O** and **Q** in the absence of alkyne-allene isomerization, as well as in the previously studied quadupolar molecules. In these cases, exciton localization is driven by the gain in solvation energy, that leads to a strong stabilization of the dipolar exciton.**

The fact that ES-SB is observed with **O** in non-polar solvents is due to the alkyne-allene isomerization. This substantial structural distortion allows for a significantly larger energy stabilization in non-polar environments that almost compensates for the loss of excitonic interaction energy.'

REVIEWERS' COMMENTS:

Reviewer #2 (Remarks to the Author):

Having read carefully the authors' responses to the reviewers' comments, I find that the authors have managed to address all concerns of the reviewers. The authors complemented the manuscript by revising the figure I pointed out and responded carefully according to our minor comments. For these reasons the manuscript is recommended for publication in "Nature Communications" without further revisions.